# Harnessing the Power of Purple Sweet Potato Color and *Myo*-Inositol to Treat Classic Galactosemia

**DOI:** 10.3390/ijms23158654

**Published:** 2022-08-04

**Authors:** Synneva Hagen-Lillevik, Joshua Johnson, Anwer Siddiqi, Jes Persinger, Gillian Hale, Kent Lai

**Affiliations:** 1Department of Pediatrics, University of Utah School of Medicine, Salt Lake City, UT 84108, USA; 2Department of Nutrition and Integrative Physiology, University of Utah, Salt Lake City, UT 84108, USA; 3Division of Reproductive Sciences, Aurora, CO 80045, USA; 4Division of Reproductive Endocrinology and Infertility, Aurora, CO 80045, USA; 5Department of Obstetrics and Gynecology, Aurora, CO 80045, USA; 6College of Medicine, University of Florida, Jacksonville, FL 32209, USA; 7Ecology and Evolutionary Biology, University of Colorado, Boulder, CO 80302, USA; 8Department of Pathology, University of Utah School of Medicine, Salt Lake City, UT 84112, USA

**Keywords:** Classic Galactosemia, Integrated Stress Response, primary ovarian insufficiency, eukaryotic initiation Factor 2 alpha (eIF2ɑ), *myo*-inositol, purple sweet potato color, antioxidant, supplements, cerebellum, hepatocyte balloon-cell change

## Abstract

Classic Galactosemia (CG) is a devastating inborn error of the metabolism caused by mutations in the GALT gene encoding the enzyme galactose-1 phosphate uridylyltransferase in galactose metabolism. Severe complications of CG include neurological impairments, growth restriction, cognitive delays, and, for most females, primary ovarian insufficiency. The absence of the GALT enzyme leads to an accumulation of aberrant galactose metabolites, which are assumed to be responsible for the sequelae. There is no treatment besides the restriction of dietary galactose, which does not halt the development of the complications; thus, additional treatments are sorely needed. Supplements have been used in other inborn errors of metabolism but are not part of the therapeutic regimen for CG. The goal of this study was to test two generally recognized as safe supplements (purple sweet potato color (PSPC) and *myo*-inositol (MI)) that may impact cellular pathways contributing to the complications in CG. Our group uses a *GalT* gene-trapped mouse model to study the pathophysiology in CG, which phenocopy many of the complications. Here we report the ability of PSPC to ameliorate dysregulation in the ovary, brain, and liver of our mutant mice as well as positive results of MI supplementation in the ovary and brain.

## 1. Introduction

Classic Galactosemia (CG) is a devastating inborn error of the metabolism caused by deleterious mutations in the gene encoding the enzyme galactose-1-phosphate uridylyltransferase (GALT). In the absence of GALT activity, an accumulation of galactose metabolites such as galactose-1 phosphate (gal1P) and galactitol, as well as deficiency of UDP-galactose, occurs in the cell leading to a variety of cellular abnormalities (Figure 1) [1]. Currently, the only treatment for CG is strict restriction of galactose from the diet; however, while dietary restriction can prevent death in the neonatal period, lifelong compliance does not protect individuals from the sequelae.

Some of the complications for individuals with CG include growth restriction, progressive development of severe neurological deficiencies (motor disorders, cognitive impairment, speech delays), and, nearly universally among females, primary ovarian insufficiency (POI) [2,3,4]. POI is a disorder of the ovary, resulting in hypergonadotropic hypogonadism, compromised ovarian function, fertility issues, and early menopause [5]. Beyond problems with fertility, the concomitant early menopause due to ovarian failure puts a female at risk for comorbidities such as cardiovascular aging, adrenal insufficiency, increased adiposity, and decreased bone mineral density [6,7,8].

Recent studies report the bulk of patients (85%) will experience some type of neurological impairment [8]. Changes in gray and white matter as well as cerebellar atrophy are also observed in the brains of Galactosemic patients, with the severity linked to clinical outcome [9,10]. While POI only affects females, the neurological impairments seen in CG affect both males and females equally [8,9]. The burden and costs of managing the neurological impairments and the psychological stress experienced by females with POI are immeasurable [11,12]. Thus, additional treatments beyond diet are sorely needed.

The use of supplements/nutraceuticals as an ancillary treatment is not new in other inborn errors of metabolism [13,14,15]. However, in CG, no supplements are ever used to target galactose induced cellular dysfunction, likely since the underlying molecular mechanisms responsible for the long-term complications are only partially understood [8]. Some of the proposed mechanisms that account for the dysfunction in CG include oxidative stress, endoplasmic reticulum stress (ER stress), and improper glycosyl/galactosylation [16,17,18,19,20,21,22]. Recent work from our lab and others have implicated the Integrated Stress Response (ISR) as a major molecular pathway involved in the pathology of CG [21,23,24,25,26], and thus manipulation of this pathway could lead to improved outcomes. The ISR is activated under stress events in the cell such as viral infection, heme deficiency, DNA damage, and ER stress [27]. Four kinases, each activated by specific stressors, phosphorylate the eukaryotic transcription initiation factor alpha (eIF2ɑ) at Serine 51 (P-eIF2ɑ), which results in stalled global protein translation, and the upregulation of genes/repair mechanisms to protect the cell from death (Figure 1) [28]. Our lab uses a *GalT* gene-trapped mouse model (*GalT*KO) to study the pathophysiology of CG [20,29,30], and we have shown impaired expression of key players in the ISR pathway in the brain [30] and the ovaries [23] of these animals when compared to their wild-type counterparts. Therefore, for the first time, our results implicated the ISR in the pathophysiology of the neurological and ovarian defects in the *GalT*KO mice.

The aim of this study was to test whether daily supplementation of two supplements/nutraceuticals (purple sweet potato color (PSPC) and *myo*-inositol (MI)), that are generally recognized as safe (GRAS), could correct the impaired ISR signaling and mitigate other suspected cellular pathologies in our mouse model. The two supplements were selected because both are GRAS, have completed human clinical trials for other diseases [31,32,33,34,35], and for their potential ability to act on aberrant molecular pathways in CG. The dark purple hue of purple sweet potatoes (PSPC) is an abundant, inexpensive source of phytonutrients called anthocyanins. Anthocyanins as a potential therapy for Classic Galactosemia was originally a hypothesis proposed by Dr. David J. Timson [36,37]. Anthocyanins are phytochemicals, which are bioactive ingredients with strong antioxidant, anticancer, and anti-inflammatory effects [38,39,40,41]. Anthocyanins have the potential to impact harrowed cellular pathways in CG by scavenging free radicals, reducing cellular stress, and supporting gene translation of antioxidant response element (ARE) pathways [41,42]. Inositols are polyols and ubiquitous substrates for a variety of cellular processes with deficiencies reported to contribute to metabolic syndromes (insulin signaling), neurological, endocrine, and ovarian diseases [43,44,45]. MI has several bioactive properties such as regulating Ca^2+^ signaling, endocrine support, substrates for cell membranes, antioxidant effects, and neurotransmitter signaling [43,46]. Supplementation with MI could impact the ISR by alleviating ER stress, supporting the ovary and brain by restoring inositol regeneration and abundance and aiding in interconnected metabolic pathways such as insulin signaling and glycolysis, pathways that may be involved in pathology of CG [31,43,44,47,48,49,50].

Here we use our mouse model of CG to show the ability of PSPC supplementation to increase the ovarian reserve, normalize molecular markers of the ISR in the ovary, correct fertility, fecundity, and improve the tissue structures of the brain and liver. In addition, we see positive effects on the ovarian and brain histology with MI supplementation. Overall, our data support the trial and use of the selected supplements as part of the treatment regimen for patients with CG.

## 2. Results

Through a variety of measures, we show here that supplementation of PSPC has a positive impact on the ovarian, brain, and liver morphology, as well as the fertility and fecundity in our mouse model of Classic Galactosemia. We also show MI supplementation impacts the ovarian reserve, restores molecular markers in the ovary, and improves the brain tissue structure in treated *GalT*KO mice.

Three groups of mice (control diet, MI, and PSPC supplemented diets) were used in this study. One group of male and female WT mice on a control diet was also used for some comparisons (ovarian immunofluorescence staining, liver and brain histology, serum AMH), with all data taken at the same age (70D) and with about an even split between male and female samples for the brain and liver comparisons. Mice were randomized to their respective diets at weaning (21D of life) and stayed on the diet ad libitum until 70D, when they were sacrificed for data collection. All mice were weighed weekly and assessed to ensure adequate dietary intake.

Characterization of the custom PSPC supplement added to the mouse diet revealed the presence of anthocyanidins and the bioactive ability of the PSPC to decrease reactive oxygen species when added to primary fibroblasts in vitro.

The purple color in purple sweet potatoes is considered a rich source of anthocyanins; the pure form is not readily available as a dietary supplement, but is primarily used as a source of natural food coloring [38]. Anthocyanins are derived by glycosylation of mainly six aglycone compounds called anthocyanidins [51]. To determine the presence of anthocyanidins in our compound, we performed HPLC with our PSPC extract. Two primary anthocyanidins were present in the PSPC extract and suspected to be delphinidin and petunidin, with peaks at 7.6 and 11.2 min, respectively (Figure 2A). To test the bioactivity of our PSPC extract, 0.1% PSPC was added to cell culture medium of *GalT*KO primary fibroblasts, and the amount of reactive oxygen species (ROS) under stressed (galactose added) conditions was measured. The addition of PSPC to cell culture medium of galactose-challenged *GalT*KO fibroblasts cells led to the decreased production of ROS by 92% (Figure 2B).

In vivo PSPC supplementation increased primordial follicles, enhanced P-eIF2ɑ fluorescence in primordial oocytes, reduced apoptosis in granulosa cells, and restored AMH to WT levels in *GalT*KO mice.

For ovarian assessment, female mice were fed a diet containing PSPC for seven weeks after weaning (to reach the age at which pairing with a male is optimal), with serum and tissue collected at the termination of the study period. For one ovary, primordial, primary, secondary/preantral, and antral follicles were counted in serial sections; the other ovary was prepared for immunofluorescence staining for ɣ-H2AX and P-eIF2ɑ. After staining, primordial follicle oocytes were identified in the DAPI channel with the fluorescence intensity measured. For ɣ-H2AX, granulosa cells with a positive stain above a determined threshold from preantral and antral follicles were counted and quantified.

*GalT*KO female mice supplemented with PSPC had significantly more primordial follicles (an average of 2385 verses 1460 per ovary) than *GalT*KO mice on control diet, but no significant differences in other types of follicles (*p* = 0.01, two-way ANOVA, Figure 3A). Serum AMH was normalized to wildtype levels at this age and was significantly less than untreated *GalT*KO mice (*p* = 0.05, one-way ANOVA, Figure 3B). PSPC supplementation increased the overall staining intensity for P-eIF2ɑ in primordial follicle oocytes (*p* = 0.008, MWU, Figure 3C). The data for P-eIF2ɑ were also plotted as a histogram frequency distribution and compared with the KS test to evaluate potentially dynamic target protein levels. The distribution for P-eIF2ɑ was significantly different between untreated and the PSPC-treated group (*p* = 0.005, KS, Figure 3D). The staining of primordial oocytes is depicted in Figure 3E. The number of positive stained preantral and antral follicle granulosa cells for ɣ-H2AX were counted and compared. PSPC-treated *GalT*KO ovaries had significantly less granulosa cells positively stained for ɣ-H2AX (on average, 59 versus 258 per ovary) than untreated *GalT*KO ovaries (*p* = 0.05, one-way ANOVA, Figure 3F), and a similar number to wildtype mice.

Female *GalT*KO mice treated with PSPC for seven weeks prior to pairing with males showed normal cycling, consistent time-to-pregnancy between litters, and significantly more pups per litter and number of litters achieved in a five-month period.

A separate group of female *GalT*KO mice were randomized to either control diet or PSPC supplemented diet for seven weeks after weaning. Two weeks prior to pairing with a male *GalT*KO mouse, estrus cycles were measured daily by vaginal cytology. Normal cycling was considered as a proportion of expected progression through the stages (estrus, metestrus, diestrus, and proestrus) at least every two days. Pairs were assessed daily for signs of pregnancy and delivery, with the number of pups and date of birth noted.

Females treated with PSPC had normal progression of estrus cycling 80% of the measured time, whereas untreated mice only 33% of the time at 56 D (*p* = 0.0062, Fisher’s exact test, Figure 4A). However, at the finish of the breeding trail (217D), there was not quite a significant difference in the number of normal cycling days for PSPC treated mice (*p* = 0.0556, Fisher’s exact test, Figure 4B). The days between litters were monitored and consistent in the number of days over five litters for PSPC-treated females; untreated females had an increasing number of days between litters after the first litter, with only one pair achieving more than two litters in the study period (*p* = 0.005 and *p* = 0.003, two-way ANOVA and linear regression, Figure 4C). The average number of pups per litter was significantly more for the PSPC-treated females (*p* < 0.0001, two-way ANOVA, Figure 4D), which was consistent over five litters.

PSPC supplementation increased dendritic arborization, restored gray matter widths to WT levels in the cerebellum, and ameliorated balloon-like cell change in the livers of treated *GalT*KO mice.

Histological sections of brain were stained for calbindin, which is present in Purkinje cells of the cerebellum. Sections of cerebellum were imaged at the same settings and evaluated for changes in morphology between the WT and the mutant mice. Twenty measurements were taken to determine the average thickness of the molecular and granular layers of three papillae, which were combined to measure gray matter in the cerebellum. WT and PSPC-treated *GalT*KO mice had overall thinner gray matter widths in the cerebellum compared to untreated *GalT*KO mice (*p* < 0.0001 and *p* < 0.0001, KW, Figure 5A). Qualitative blinded comparison between the groups revealed intricate arborization in the molecular layer of the cerebellum in PSPC-treated mice compared to thin, straight branches in the untreated *GalT*KO samples (Figure 5B).

Evidence of fibrosis and cell morphology in the liver was evaluated. There was no sign of fibrosis in any liver sample using the Batts-Ludwig scoring (0–4) with Masson’s trichrome staining. However, there was significant balloon cell change of hepatocytes in the midzonal regions, which was assessed in both groups in a blinded manner and scored as high (≥3) or low (<3). While 80% of the PSPC-treated livers assessed scored “low” for balloon cell change compared to 66% of the untreated *GalT*KO livers, it was not statistically significant (*p* > 0.9999, Fisher’s exact test). Representative images are shown in Figure 5C.

*Myo*-inositol supplementation increased the number of ovarian primordial follicles and decreased granulosa cell apoptosis in treated mutant mice.

After seven weeks of MI supplementation, mice were sacrificed, with serum and ovaries collected and prepared as described above.

MI supplemented mice, on average, had 2253 primordial follicles per ovary compared to 1460 in the untreated *GalT*KO mice (*p* = 0.05, two-way ANOVA, Figure 6A). There was not a significant difference in the other types of follicles measured. AMH levels were compared at 70D between WT, *GalT*KO mice on control diet, and MI treated mice. AMH was higher for both *GalT*KO and MI treated *GalT*KO mice than WT (*p* = 0.01 and *p* = 0.05, one-way ANOVA, Figure 6B), and therefore MI treatment did not normalize AMH to WT levels. Overall fluorescence intensity of P-eIF2ɑ in primordial follicle oocytes was lower in *GalT*KO ovaries than in MI supplemented *GalT*KO ovaries (*p* = 0.008, MWU, Figure 6C). The distribution of staining intensity was also different between *GalT*KO and MI treated *GalT*KO ovaries (*p* = 0.005, KS, Figure 6D). In preantral and antral follicles, untreated *GalT*KO ovaries had a significantly greater number of positively stained granulosa cells for ɣ-H2AX than WT; MI supplementation did not decrease the number of granulosa cells positive for ɣ-H2AX (*p* = 0.3287, one-way ANOVA, Figure 6E (image) and Figure 6F).

Supplementation with MI resulted in decreased concentrations of two unidentified inositol isoforms in the cerebella of treated mutant mice and improved dendritic arborization; however, liver balloon-cell change was not mitigated.

*Myo*-inositol, inositol phosphate, and two other unidentified inositol isoforms were measured in the cerebella of untreated and MI supplemented *GalT*KO mice. Supplementation of MI did increase the concentration of *myo*-inositol in the treated mutant cerebella compared to untreated mutants, but with the number of animals we used for this pilot study, the difference was deemed insignificant *(**p* = 0.0518, *t* test, Figure 7A). Inositol phosphate concentrations in the cerebellum were not changed after MI supplementation. However, there was a significant decrease in two undetermined inositol species in the cerebella after MI supplementation (*p* = 0.01 and *p* = 0.006, *t* test, Figure 7A).

As above, sagittal histological sections of brain were stained for calbindin, and the morphology was compared. Similar results to the PSPC supplemented mice, MI treated *GalT*KO mice had overall thinner gray matter compared to untreated *GalT*KO mice (*p* < 0.0001 and *p* < 0.0001, KW, Figure 7B). Qualitative blinded comparison between the three groups revealed dramatic dendritic arborization of MI supplemented cerebellum molecular layers compared to untreated *GalT*KO samples (Figure 7C).

Livers were analyzed in the same way as the PSPC supplemented group for evidence of fibrosis and balloon-cell change between untreated *GalT*KO mice and MI treated *GalT*KO mice. Again, there was also no evidence of fibrosis in any liver assessed. Fisher’s exact test was used to determine whether *GalT*KO mice supplemented with MI had a higher chance of receiving a low score (<3) for balloon cell change, which was not significant (*p* > 0.999, Fischer’s exact test, representative image—Figure 7D).

## 3. Discussion

While lifesaving in the neonatal period, the dietary restriction of galactose to treat CG falls short of preventing devastating complications such as POI, neurological impairments, and cognitive dysfunction. Despite many years of study, the exact cellular mechanisms contributing to the sequelae remain elusive, but are now being shown as multi-factorial. Our lab and others have demonstrated the involvement of the ISR, oxidative stress, the canonical PI3K/AKT signaling, and inflammatory pathways as contributing to the pathophysiology [16,20,21,22,25,26,52,53]. We show here that PSPC supplementation, suspected to target the ISR and oxidative stress involved in CG, functions to improve the ovarian, brain, and liver histology while also improving fertility and fecundity in our mouse model of CG. MI supplementation also revealed positive support of the ovary and brain in treated mutant mice.

The phytonutrients in purple sweet potatoes, called anthocyanins, are thought to be responsible for their bioactivity. Anthocyanins are derived by glycosylation or acylation of primarily six anthocyanidin backbones (e.g., delphinidin and petunidin) [51]. Anthocyanins are absorbed in the stomach and small intestine and metabolized further by the gut microbiome [54]. The powerful antioxidant and anti-inflammatory properties of the anthocyanins in the PSPC is the assumed mechanism to relieve cellular and oxidative stress [55]. The cellular generation of free-radicals (called oxidative stress) and their mitigation by the body’s natural antioxidant defense is a normal process of human metabolism; however, increased oxidative stress is associated with a variety of chronic diseases including neurogenerative disorders [56] and ovarian dysfunction [57,58,59], and is present in models of CG [22,60]. Rodents fed high amounts of galactose to elicit galactosemia showed vast improvement in cognition and hepatocellular injury and decreases in oxidative and ER stress after supplementation with PSPC [41,61,62]. While PSPC has not been trialed specifically for ovarian failure, other powerful antioxidants have [63,64]. Additionally, sufficient antioxidant balance is crucial for follicle and oocyte development [65].

Deficiency of MI is a long-suspected mechanism of cellular distress in CG, with evidence of decreased MI in the brains of deceased patients [48,66]. MI is synthesized endogenously or is obtained by the diet [31,49]. Gal1P is a known inhibitor of inositol monophosphatase (IMPase), which is an important enzyme for the regeneration of cellular MI [1,21,48]. High levels of galactose and galactitol in CG are correlated with poor MI transport into the cell, possibly also contributing to less available inositol [31,47]. Reduced cellular MI can interfere with Ca^2+^ signaling [43,67], which elicits an ER stress response, and thus an activated ISR [68]. Supplementation of MI and its isoforms has been used and studied for polycystic ovarian syndrome with some success in helping ovarian and endocrine functioning [44,69,70,71]. In the brain, MI is abundant and can serve to maintain osmotic balance and support normal Ca^2+^ signaling in dendrites [49,72]; altered levels of inositol are associated with degenerative cognitive disorders and supplementation has been shown to alleviate some mental illnesses [43,45,50].

In our study, supplementation with PSPC led to a greater number of primordial follicles and increased fluorescent staining intensity of P-eIF2ɑ in those primordial follicle oocytes, indicating proper functioning ISR [25]. Increased P-eIF2ɑ is associated with inhibited growth and increased repair, and thus primordial follicle dormancy and healthy oocytes [25,27,73]. The number of primordial follicles is considered the ovarian reserve as arrested primordial follicles become activated (a process termed folliculogenesis) and grow into primary follicles, secondary follicles, and, eventually, antral and Graafian follicles to ovulate an oocyte. However, most follicles beyond the primary stage will perish through a process called atresia. The growth of primordial follicles to the fate of ovulatory or atretic follicles is a continuous process throughout a female’s reproductive life; thus, the appropriate timing of growth and number of primordial follicles is integral in preserving ovarian function until anticipated menopause.

The mechanism of primordial follicle preservation is likely reduced cellular stress through ROS scavenging, antioxidant support, and proper DNA damage repair, aided by the anthocyanin action in PSPC [42,55]. However, additional molecular studies to determine the pharmacodynamics of PSPC are needed, such as measuring redox status in various tissues and circulating anthocyanin metabolites. We saw fewer secondary follicle granulosa cells positively stained for ɣ-H2AX, suggesting less follicle atresia and growth. The decreased serum AMH of PSPC supplemented mutant mice to WT levels supports decelerated growth of secondary follicles, and, while not statistically significant, follicle counts showed fewer growing follicles in the PSPC supplemented ovaries than untreated. The positive results of the breeding trial (normal number of days between pregnancies, more litters in the trial period, and twice the number of pups per litter) after PSPC supplementation were perhaps the most striking for the efficacy of PSPC in our mutant mice. We saw the same improvement in primordial follicle numbers and fertility and fecundity with our previous studies using Salubrinal [23], which is a compound that acts to preserve the phosphorylated state of eIF2ɑ [74]. Beyond acting to keep eIF2ɑ phosphorylated in the primordial oocyte, the exact mechanism of supporting the ovary, fertility, and fecundity is unknown.

MI supplementation also resulted in an increased number of primordial follicles and increased fluorescent intensity of P-eIF2ɑ in primordial oocytes, suggesting it also may support a functioning ISR in the ovary [25]. In contrast to PSPC, a similar number of secondary follicle granulosa cells stained positive for ɣ-H2AX in MI treated ovaries as untreated mutants; granulosa cells positive for ɣ-H2AX can be a marker of atresia in the context of growing secondary follicle granulosa cells. AMH was not reduced to WT levels after MI supplementation, and although not statistically significant, the number of growing follicles was slightly higher in MI treated *GalT*KO mice than untreated *GalT*KO mice. As more growing follicles could correspond to a higher AMH, there is evidence of accelerated secondary follicle growth, which we showed as a mechanism of follicle burnout and POI in untreated *GalT*KO mice compared to WT, previously [75]. A breeding trial with this cohort would confirm the support of MI supplementation in the primordial follicles and whether improvement in fertility and fecundity is possible.

The cerebella of WT and both MI and PSPC-treated groups of mutant mice revealed differences in gray matter thickness and dendritic arborization patterns compared to untreated *GalT*KO mice. In a study with human patients, increases in gray matter and decreases in white matter were reported in individuals with CG compared to controls in the cortex of the brain; however, there were no differences in the cerebella of the participants [9]. Cellular mechanisms such as apoptosis/autophagy and ER stress are tightly regulated during cerebellar development and crucial for the proper functioning of neuronal cells. Apoptosis is part of normal development and can act to prune superfluous neurons, while autophagy is involved in neuronal reconstruction and elicits proper morphology [76]. Appropriate functioning of ER stress pathways also appears to be crucial for the development of the cerebellum. Mutations in the ER stress protein BiP, which is involved in the sequestering of misfolded proteins in the ER, led to accumulation of ubiquitinated proteins in Purkinje cells and, thus, neurodegeneration [77]. Chronic ER stress can also lead to Purkinje cell loss due to disruptions in calcium signaling [78]. In this study, the thicker gray matter in the untreated *GalT*KO cerebellums could possibly be a result of improper apoptosis and autophagy, leading to an accumulation of ubiquitinated proteins and extra Purkinje cells at the young age of 70D. Eventually, this stress could lead to loss of Purkinje cells in older mice, as our lab showed previously [29]. It could also be a result of osmotic swelling at this stage of early development, which is implicated in CG [3,48,66]. While not statistically significant, the amount of *myo*-inositol in the treated mutant cerebella was higher than in untreated mutants. Measuring inositol levels in other tissues after supplementation would give a clearer picture of the pharmacokinetics of MI in our mouse model.

The dendrite arborization patterns also differed between our untreated *GalT*KO mice and the two supplemented mutant groups. In untreated mutant mice, the dendrites appeared punctuated and had irregular staining for calbindin compared to the two treatment groups and WT. This pattern of arborization is present in mouse models of prion disease, where misfolded proteins and ER stress are involved in the pathogenesis [79]. Our characterization here is quite limited to the basic morphological comparisons of the cerebellum between the treatment groups. To understand the true mechanism of dysfunction in the *GalT*KO cerebellum and how PSPC and MI supplementation act on these mechanisms, more work is needed to determine the significance of the differences reported here.

While there was no evidence of fibrosis in the liver of any group assessed, there was apparent balloon-like cell change in the untreated *GalT*KO and MI treated *GalT*KO livers, but not in the PSPC supplementation group. Other models of aberrant metabolism, mainly non-alcoholic fatty liver disease (NAFLD), show similar morphological changes in addition to accumulation of lipid in hepatocytes with over-nutrition [80]. Balloon-like changes in hepatocytes can occur in conditions of oxidative stress and misfolded proteins, as seen in ER stress and metabolic syndrome [81]. In support of PSPC supplementation, antioxidants are helpful in ameliorating liver inflammation and damage, as was seen in a human trial with purple sweet potato juice [35] and many other models of metabolic dysfunction [80,82,83,84]; therefore, it is likely the PSPC supplement impacted the hepatocytes directly as an antioxidant. Additional molecular studies are needed to determine the liver involvement in not only our *GalT*KO mouse model but also with the dietary treatments.

This study contains some caveats. We did not perform an increasing dose response in the mice with our PSPC and MI supplements, which would be necessary to determine the smallest effective dose. The human equivalent dose (HED) was calculated (See Materials and Methods, Custom Diets), but it is unclear if the amount we supplemented in the mice would elicit a positive response in humans. Animal and human studies have not shown a toxic level of anthocyanin intake, therefore, a trial in humans may prove safe and potentially effective [85]. For MI, human studies have supplemented MI up to 12 g per day before seeing adverse effects; at this amount, diarrhea and gastrointestinal upset have been reported, which warrants caution when using with patients [69]. The presence of anthocyanidin components in our PSPC extract was confirmed, but further work is needed to determine the specific species in the compound; quantifying the anthocyanidin and glycosylated anthocyanin compounds with mass spectrometry would give us a better understanding of the potential mechanism of action, absorption, and metabolism of the compounds present in the PSPC [51]. Additionally, this study provided a brief characterization of the positive effects of PSPC and MI on our mouse model. Future work will focus more on the molecular mechanisms and pathways affected by PSPC and MI supplementation, especially in the brain and liver.

## 4. Materials and Methods

### 4.1. GalT Gene-Trapped (GalTKO) Mouse Model

Mutant *GalT*KO and wildtype (WT) mice used in this study were established as previously described [60], with the genetic background of all mice confirmed by Transnetyx, Cordova, TN, USA using real-time PCR. The animal studies were conducted and approved by the guidelines outlined for the care and use of laboratory animals by the University of Utah Institutional Animal Care and Use Committee (IACUC, protocol# 21-05004, approved 18 May 2021).

### 4.2. Custom Diets

All specialized diets (including control) were supplied by TestDiet^®^ (Richmond, IN, USA). MI was purchased from Sigma-Aldrich Inc. (St. Louis, MO, USA, cat#I5125). The anthocyanidin rich-purple-colored powder, PSPC (90% PSPC and 10% citric acid) was custom extracted from Stokes Purple^®^ variety purple sweet potatoes by Food Ingredient Solutions, LLC (Teterboro, NJ, USA). Both supplements were sent to TestDiet^®^ and specialty added to the control diet as a supplement, processed and pelleted with minimal heat to preserve the compounds. The amount of PSPC (0.28% or 700 mg/kg/d) and MI (1.2% or 3 g/kg/d) added to the diet was determined by previous studies using the supplements [62,69,86] and calculated by the estimated intake of 3–5 g of food per day per mouse weighing 20 g. The relative amounts of anthocyanidin in the PSPC compound were measured by HPLC-MS techniques. The human equivalent dose (HED) calculations of each supplement were conducted with the following equation [86] and determined to be 49.8 mg/kg for PSPC and 213 mg/kg for MI:HED (mg/kg = Animal NOAEL mg/kg) × (Weight_animal_ [kg]/Weight_human_ [kg])^(1−0.67)^

### 4.3. Hormone Analysis

Serum Anti-Müllerian hormone (AMH) was collected as an indirect measurement of secondary follicle growth and was evaluated by The University of Virginia Center for Research in Reproduction Ligand Assay and Analysis Core (Charlottesville, VA, USA).

### 4.4. Primary Fibroblast and Cell Culture Conditions

Skin from age- and sex-matched WT and *GalT*KO mice was harvested with primary fibroblasts isolated using previously described techniques [30]. Fibroblasts were then propagated in Dulbecco’s Modified Eagle Medium (DMEM; ThermoFisher, Waltham, MA, USA, cat# 11965092, cat# 12320032, and cat#11966025), which was supplemented with 12% fetal bovine serum (FBS), 1% penicillin, and 0.5% streptomycin at 37 °C and 5% ambient CO_2_. To assess reactive oxygen species (ROS), *GalT*KO fibroblasts were plated in low-glucose (1 g/L) medium for 24 h. The medium was changed to glucose-free medium (supplemented with 1.1% sodium pyruvate; ThermoFisher, Waltham, MA, USA, cat# 11360070), and 0.05% galactose was added to the medium. Cells were treated with 100 µM TBHP as a positive control, 0.1% PSPC extract, 0.26 mM, 0.52 mM, and 1.04 mM citric acid to account for the citric acid present in the PSPC extract, which was equivalent to 0.52 mM. After treatment, the DCFDA/H2DCFDA-Cellular ROS Assay Kit (abcam, Waltham, MA, USA, cat# ab113851) was performed according to protocol and fluorescence measured with a 96-well microplate reader.

### 4.5. Immunofluorescence Studies

At postnatal day 70, ovaries and the brain were extracted from *GalT*KO mice as described previously [29] and placed in 10% non-buffered formalin for 12 and 72 h, respectively. The tissues were re-hydrated with 70% alcohol and embedded in paraffin. The Research Histology Core Facility of the ARUP Laboratories (Salt Lake City, UT, USA) sectioned all tissues. Two cross sections of each tissue, five microns thick, were placed on each slide, one for a secondary antibody-only control. For the brain, the organ was cut and embedded on the sagittal plane, thus, the cut sections represent a sagittal view. Cross sections were used to localize the expression of P-eIF2ɑ (Serine 51) (1:100; Cell Signaling Technologies, Danvers, MA, USA, cat# 3398) and phospho-histone H2A.X (Ser139) (ɣ-H2AX) (1:480; Cell Signaling Technologies, Danvers, MA, USA, cat# 9718S) in the ovary and calbindin (D1I4Q) XP^®^ (1:800; Cell Signaling Technologies, Danvers, MA, USA, cat#13176) in the brain (cerebellum).

Antigen retrieval was performed by incubating sections in either 10 mM citrate buffer (pH 6.0) or 1× EDTA buffer (pH 8.0) at 120 °C for 20 min after deparaffinization and rehydration. Then, blocking was completed with 1% (*w*/*v*) bovine serum albumin plus 10% normal goat serum at 21 °C for 60 min. Sections were incubated overnight in a humidified chamber with the selected primary antibodies at 4 °C. After washes with phosphate buffered saline (PBS) (pH 7.4), sections were incubated for 60 min at room temperature with Alexa Fluor 647 (1:500; abcam, Danvers, MA, USA, cat# ab150075) secondary antibodies. Nuclei were counterstained with 300 µM DAPI (ThermoFisher, Waltham, MA, USA, cat# D1306) for four minutes. After DAPI, 20× TrueBlack (Biotium, ThermoFisher, Waltham, MA, USA, cat#NC1125051) diluted to 1× in 70% ethanol was applied to all sections for 30 s to reduce autofluorescence, then rinsed with PBS. Slides were mounted with EMS mounting medium (Electron Microscopy Sciences, Hatfield, PA, USA, cat# 17989-50), with covers applied and held in place with clear nail polish. Secondary antibody only sections were subjected to the same procedures, except blocking buffer only was added in place of the primary antibody for the overnight incubation. Imaging was performed with a Nikon Widefield microscope with 20× lenses (The University of Utah Cell Imaging Core, Salt Lake City, UT, USA). All ovary sections were imaged using the same camera settings across WT and *GalT*KO samples for each antibody.

For analysis, mean background autofluorescence from secondary antibody-only controls was subtracted from primary antibody-stained samples using NIS-Elements Analysis software (Nikon Corporation, Americas, Melville, NY, USA). For ovaries, follicle and cell types were sorted in the DAPI channel by their morphology, with the oocytes and granulosa cells analyzed separately for each follicle type. Oocytes were differentiated from granulosa cells by separate regions of interest in follicles where visible nuclei were present. Staining intensities were measured in the oocytes of primordial follicles. For ɣ-H2AX in secondary follicle granulosa cells, individual stained cells were counted manually after threshold settings applied instead of intensities averaged. For the average fluorescent intensities calculated, the signal was quantified and divided by the area for each region of interest (area of the oocyte).

For the brain, the cerebellum was imaged separately from the cortex at 20×. Twenty measurements of the molecular layer and the granular layer were taken for three papillae in the inferior semilunar lobules (X, IX, VIII) of the cerebellum, also using measurement tools in NIS-Elements Analysis software (Nikon Corporation, Americas, Melville, NY, USA). Measurements from these two layers were combined and compared as cerebellar gray matter. Qualitative analysis of arborization was visualized and corroborated by three separate board-certified pathologists.

### 4.6. Follicle Counts

One ovary from separate animals at 70D was harvested and fixed in Dietrich’s fixative for 12 h then rehydrated and embedded as described above. Pole-to-pole serial sections of 5 microns were cut and placed on slides by the Research Histology Core Facility of the ARUP Laboratories (Salt Lake City, UT, USA). Sections were then stained with Weigert’s Iron Hematoxylin (Electron Microscopy Services, Hatfield, PA, USA, cat# 26044-05) and Picric Acid (Electron Microscopy Services, Hatfield, PA, USA, cat# 26853-07)-Methyl blue (ThermoFisher, Waltham, MA, USA, cat#H37721-21) with cover slides mounted using Permount (Fischer Scientific, Waltham, MA, USA, cat# SP15-100). Each sample was blinded before counting and histomorphometric evaluation [87]. The methods are described briefly as follows: Intact follicles (primordial, primary, small/large preantral, and antral) were counted in every fifth section, with the total numbers estimated by multiplying by five. After evaluation, ovaries were decoded and data analyzed as untreated mutant verses treated mutants.

### 4.7. Breeding Trials

To test for improvement in the subfertility phenotype after supplementation with PSPC, two groups of female *GalT*KO mice were supplemented with a control diet or PSPC supplemented diet for seven weeks after weaning. At eight weeks of life, estrus cycles were measured daily for two weeks by vaginal cytology [88]. Females from each group were then paired at 70D of life with a control diet treated *GalT*KO male mouse and continuously mated for five litters. Serum was collected prior to pairing with a male, and Anti-Müllerian hormone (AMH) was evaluated by The University of Virginia Center for Research in Reproduction Ligand Assay and Analysis Core (Charlottesville, VA, USA). The time to pregnancy (days between litters) and the number of pups born per littler were counted for each pair and for each litter.

At 187 days of life (after 5 months of pairing), the females in both groups were separated from the males to prevent future pregnancies. Then, one month later, when females showed no signs of pregnancy, estrus cycles by vaginal cytology were again measured daily for ten days to assess cycling.

### 4.8. Liver Histology

Liver tissue was extracted from the animals at sacrifice, fixed in 10% non-buffered formalin for 48 h, and embedded as described above. Two sections of 5 microns in thickness were cut and placed on slides and stained with either Masson’s trichrome or hematoxylin and eosin (H&E) by the Research Histology Core Facility of the ARUP Laboratories (Salt Lake City, UT, USA). Blinded liver morphology was evaluated and confirmed by a board-certified pathologist. Batts-Ludwig scoring from 0 to 4 was used to assess for fibrosis [89]. For balloon-cell change score, a similar qualitative scale of 0–4 (0 = none, 4 = abundant change) was used for scoring.

### 4.9. GC-MS Quantification of Myo-Inositol in the Cerebellum

Metabolomics analysis was performed at the Metabolomics Core Facility at the University of Utah (Salt Lake City, UT, USA). Brain tissue was extracted as previously described, cut on the sagittal plane with the cerebellum separated. Tissues were flash frozen in liquid nitrogen and stored at –80 °C until extraction. For metabolite extraction, each sample was transferred to 2.0 mL ceramic bead mill tubes (Qiagen, Germantown, MD, USA, Catalog Number 13116-50). To each sample, 450 μL of cold 90% methanol (MeOH) solution containing the internal standard d4-succinic acid (Sigma, St. Louis, MO, USA, cat# 293075) was added for every 25 mg of tissue. The samples were then homogenized in an OMNI Bead Ruptor 24. After, homogenized samples were incubated at –20 °C for 1 h. After incubation, the samples were centrifuged at 20,000× *g* for 10 min at 4 °C. Then, 400 μL of supernatant was transferred from each bead mill tube into a labeled, fresh microcentrifuge tube. Another internal standard, d27-myristic acid, was then added to each sample. Pooled quality control samples were made by removing a fraction of collected supernatant from each sample. Process blanks were made using only extraction solvent, and these went through the same process steps as the samples. Everything was then dried en vacuo.

All GC-MS analysis was performed with an Agilent 5977b GC-MS MSD-HES fit with an Agilent 7693A automatic liquid sampler (Santa Clara, CA, USA). Dried samples were suspended in 40 µL of 40 mg/mL O-methoxylamine hydrochloride (MOX) (MP Bio, Irvine, CA, USA, #155405) in dry pyridine (EMD Millipore, St. Louis, MO, USA, #PX2012-7) and incubated for one hour at 37 °C in a sand bath. Twenty-five µL of this solution was added to auto sampler vials. Sixty µL of *N*-methyl-*N*-trimethylsilyltrifluoracetamide (MSTFA with 1% TMCS, ThermoFisher, Waltham, MA, USA, #TS48913) was added automatically via the auto sampler and incubated for 30 min at 37 °C. After incubation, samples were vortexed, and 1 µL of the prepared sample was injected into the gas chromatograph inlet in the split mode with the inlet temperature held at 250 °C. A 10:1 split ratio was used for analysis for most metabolites. Any metabolites that saturated the instrument at the 10:1 split were analyzed at a 150:1 split ratio. The gas chromatograph had an initial temperature of 60 °C for one minute followed by a 10 °C/min ramp to 325 °C and a hold time of 10 min. A 30-m Agilent Zorbax DB-5MS with 10 m Duraguard capillary column was employed for chromatographic separation. Helium was used as the carrier gas at a rate of 1 mL/min (Santa Clara, CA, USA).

Data were collected and analyzed using MassHunter software (Agilent, Santa Clara, CA, USA). Metabolites were identified and their peak area was recorded using MassHunter Quant. These data were transferred to an Excel spread sheet (Microsoft, Redmond, WA, USA). Metabolite identity was established using a combination of an in-house metabolite library developed using pure purchased standards, the NIST library, and the Fiehn library.

### 4.10. High-Performance Liquid Chromatography (HPLC)

To detect the presence of anthocyanidins in the PSPC sample, HPLC analysis was conducted following Harborne [90] with modifications [91]. PSPC extract was placed in 2 mL tubes with 1 mL 2 N HCL, vortexed, and soaked overnight. Samples were then centrifuged (5 min at 13,000 rpm) to pellet the PSPC sample debris, and the supernatants (~1 mL) were decanted into a new 2 mL tube. To cleave sugar moieties from anthocyanin molecules, the extract was placed in a 100–104 °C heat block for 90 min. Samples were removed and cooled to 21 °C. Retained supernatants were washed twice with 400 µL of ethyl acetate, vortexed, then centrifuged for 1 min at 13,000 rpm to restore phase separation. After removal of the ethyl acetate layer, the tubes were placed in a SpeedVac for 20 min to ensure all residual ethyl acetate evaporated. To remove remaining HCl, 150 microliters of iso-amyl alcohol was then added to the tubes, and the solution was vortexed and centrifuged for 1 min at 10,000 rpm. The iso-amyl alcohol layer containing the anthocyanins was carefully moved to a new 2 mL tube. The iso-amyl alcohol step was repeated a second time for a total of 300 µL iso-amyl extract in the new 2 mL tube. Tubes were left to dry overnight in the SpeedVac to reduce the product to an anthocyanin pellet. Extracts were eluted in 100 µL of 0.5% HCl in MeOH buffer and further diluted to a 1:50 dilution with the buffer, before analyses. Diluted extracts were injected into an Agilent 1260 Infinity system (ThermoFisher, Waltham, MA, USA). Delphinidin chloride, cyanidin chloride, peonidin chloride, malvidin chloride, and petunidin chloride were used as anthocyanidin standards. A vial of combined standards (1 µL each anthocyanidin) was brought to a 1:10 dilution with buffer. Pigments were separated using a 100-4.6 mm Chromalith Performance column (Agilent Technologies, Santa Clara, CA, USA) following a gradient elution at 30 °C using solvents A (HPLC-grade water, 0.1% trifluoroacetic acid) and C (MeOH, 0.5% HCl), with the following program: 15% C from 0 to 4 min; linear increase to 20% C from 4 to 10 min; 20% C from 10 to 14 min; linear increase to 22.5% C from 14 to 16 min; instantaneous increase to 27.5% C; 27.5% C from 16 to 18 min; instantaneous decrease to 15% C; 15% C from 18 to 21 min. Separated pigments were detected using a UV-vis Diode Array Detector coupled to the HPLC and set to 520 nm. A blank sample (buffer) was run in between every sample to wash the injection needle and to avoid contamination. Results were analyzed using the Agilent Chemstation software, and peaks in the samples were compared with those in the standards.

### 4.11. Statistical Analysis

Statistical significance was determined as indicated in results, by using one- and two-way ANOVA, repeated measures ANOVA, simple linear regression, Fisher’s exact test, and *t* tests. Non-parametric tests were used (Mann–Whitney U (MWU), Kruskal–Wallis (KW)) for not normally distributed data and the Kolmogorov–Smirnov (KS) test for primordial follicle staining distributions. The statistical program R was used to analyze and graph signal intensity distributions with KS test. GraphPad Prism version 9.2.0 was used for all other statistical tests and graphing. A *p*-value of less than 0.05 was considered statistically significant.

## 5. Conclusions

Using natural supplements in CG could be a cost-effective adjunct therapy for individuals to help prevent long-term complications. With PSPC supplementation, we see positive results in the ovary, liver, and brain, as well as improved fertility and fecundity in treated *GalT*KO mice. MI supplementation shows promising results as well in supporting the ovary and brain. The results between the two supplements seem tissue specific, where the PSPC supplementation perhaps resulted in greater improvement for the liver and ovarian function, and the MI supplementation led to greater change in the cerebellar morphology. While additional therapies are under review to treat CG, such as small molecule inhibitors [92,93] and gene/mRNA therapy [94,95,96,97], the adjunct use of natural supplements is appealing as they can be taken at home, are non-invasive, and can be stopped at any time. However, the use of these supplements in humans still requires additional study for safety and efficacy, and could be a good option for adjunct treatment for CG in the future.

## Figures and Tables

**Figure 1 ijms-23-08654-f001:**
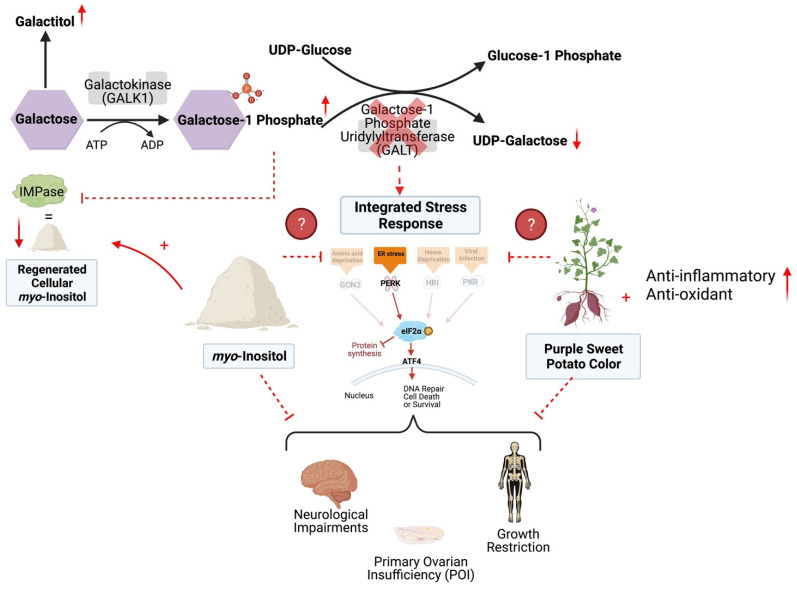
The Leloir pathway of galactose metabolism and proposed mechanism of *myo*-inositol (MI) and purple sweet potato color (PSPC) support in Classic Galactosemia. With the deficiency of GALT, galactose-1 phosphate and galactitol are present with a deficiency of UDP-galactose. This contributes to a dysregulated Integrated Stress Response (ISR), cellular stress, and DNA damage. PSPC has the potential to relieve cellular distress by bolstering the antioxidant defense, scavenging free radicals, reducing ER stress, and thus supporting the ISR; additional anti-inflammatory and antioxidant pathways are likely involved. Supplemental MI may ameliorate cellular distress by restoring levels of inositol and reducing ER stress. Supplementation of either may improve the sequelae of Classic Galactosemia. Solid arrows and inhibitor lines represent known metabolic processes; dashed red lines represent potentially inhibited metabolic processes; small solid red arrows indicate the increases or decreases of selected metabolites in the pathway. Finally, question marks with dashed lines indicate that the mechanism of the two selected supplement’s impacts on the ISR are unknown. This figure was created in BioRender.com.

**Figure 2 ijms-23-08654-f002:**
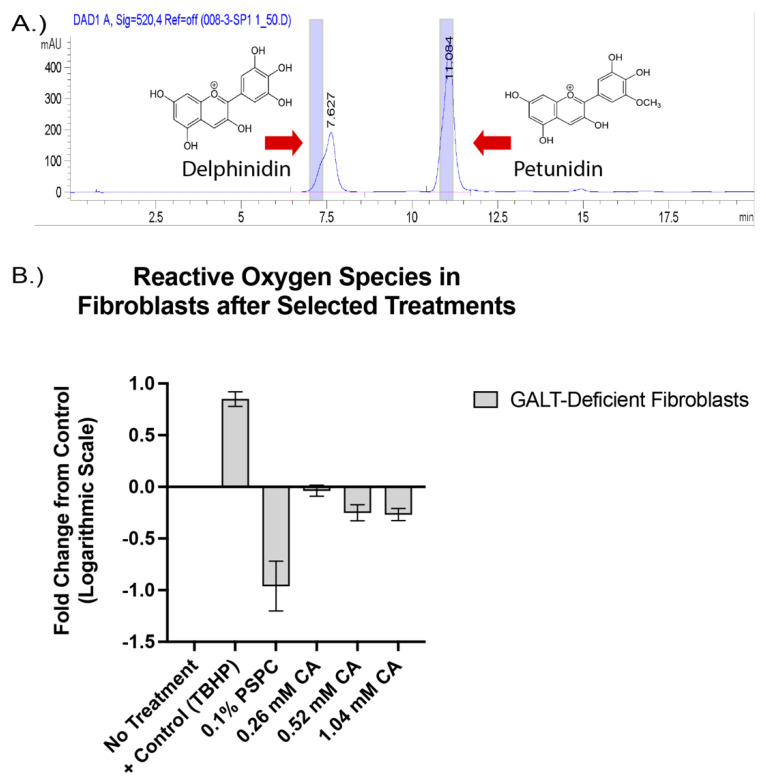
The presence of anthocyanidin compounds and bioactive properties of the purple sweet potato color (PSPC). Peak values of present anthocyanidin backbones are shown in (**A**), with blue shaded regions indicating the standard peaks. Fold change in reactive oxygen species compared to untreated *GalT*KO fibroblasts (**B**). Bars represent standard deviation in replicates. Positive control (TBHP), 0.1% PSPC, and increasing concentrations of citric acid (0.26 mM, 0.52 mM, and 1.04 mM) to account for the citric acid present in the PSPC extract.

**Figure 3 ijms-23-08654-f003:**
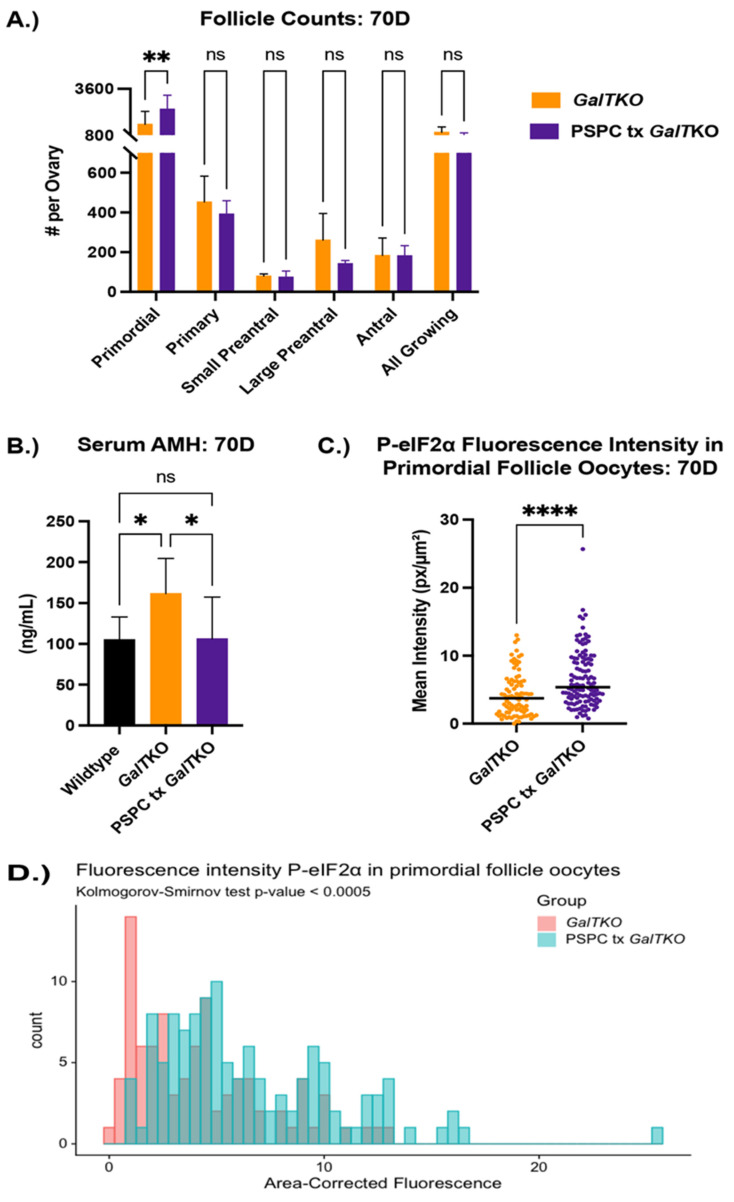
Impact of PSPC supplementation on the ovary. Follicle counts (**A**) at 70D between untreated *GalT*KO and PSPC supplemented *GalT*KO mice. Serum AMH compared between WT, *GalT*KO, and PSPC treated *GalT*KO mice (**B**). Median fluorescence intensity of P-eIF2ɑ (**C**) in primordial follicle oocytes. The distribution of primordial follicle oocytes is shown in (**D**). Representative image of primordial oocytes stained for P-eIF2ɑ, with white dashed circles showing outline of oocyte (**E**). The number of secondary follicle granulosa cells stained positive for ɣ-H2AX between WT and the two *GalT*KO groups (**F**). Data were analyzed with one- or two-way ANOVA, MWU test, or KS test. * *p* ≤ 0.05, ** *p* ≤ 0.01, **** *p* ≤ 0.0001.

**Figure 4 ijms-23-08654-f004:**
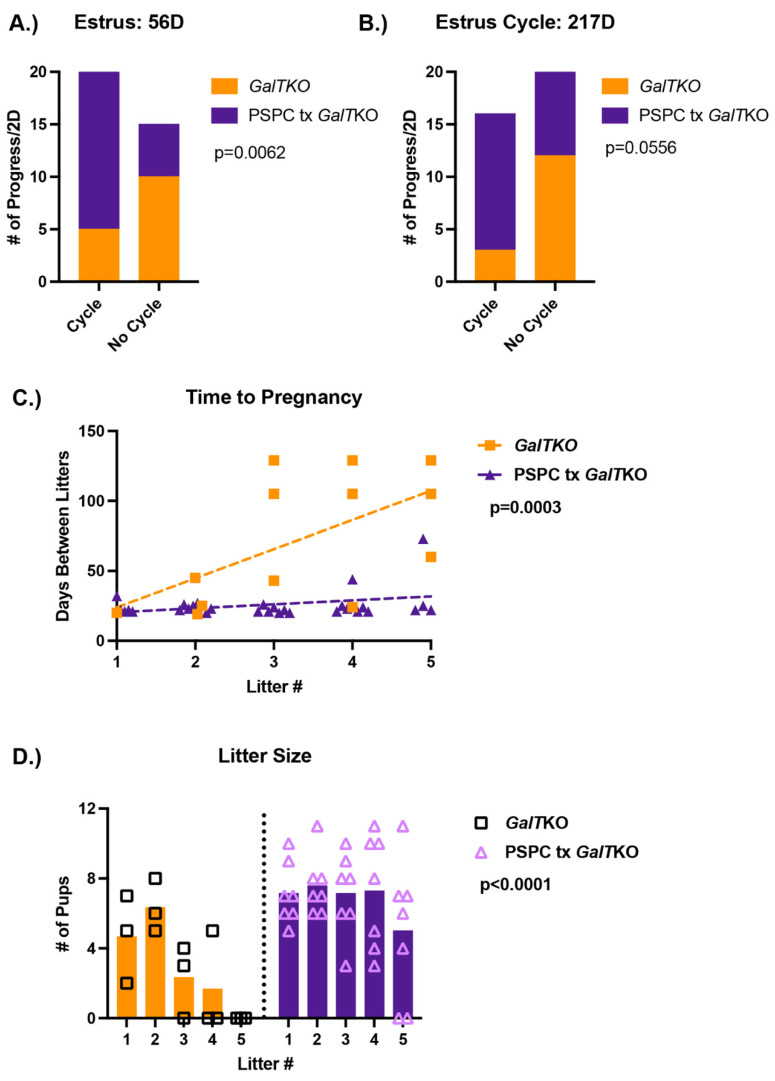
Results of the breeding trial following PSPC supplementation. Estrus cycles before pairing with a male (**A**) and following the completion of the breeding trial (**B**) (*p*-value shown from Fisher’s exact test). The number of days between litters (time to pregnancy) over five litters (**C**); *p*-value represents the difference in slope between the two groups (simple linear regression). The number of pups or litter size for untreated *GalT*KO pairs and PSPC treated *GalT*KO pairs over five litters (**D**) (*p*-value for two-way ANOVA).

**Figure 5 ijms-23-08654-f005:**
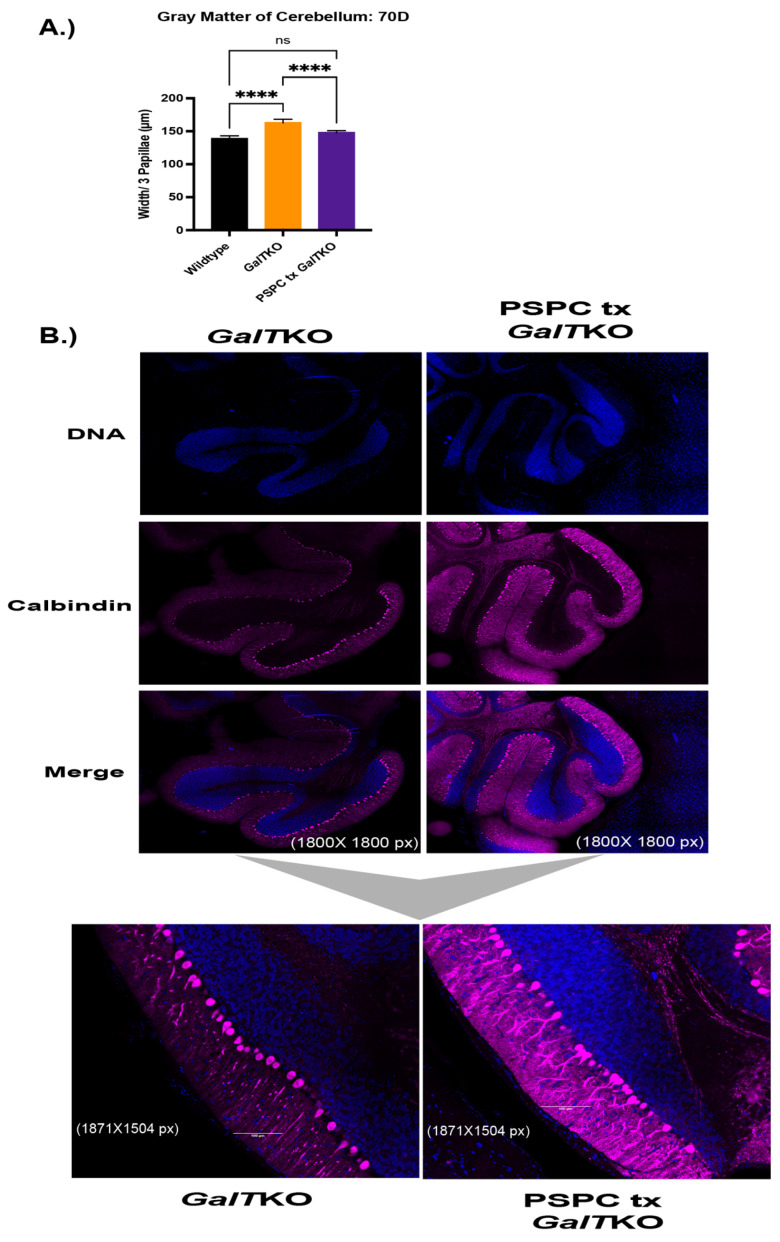
Gray matter widths, dendritic arborization, and liver histology after PSPC supplementation. (**A**) Shows gray matter thickness of treated mice compared to wildtype and untreated *GalT*KO mice. In (**B**), the first six panels are show at 1800 × 1800 pixel magnification with bottom subfigures at 1871 × 1504 pixel magnification. (**C**) Panels on the left show H&E-stained liver histology at 10× and 40× in the subfigure (**C**). Black arrows depict balloon-cell change areas in hepatocyte cytoplasm. **** *p* ≤ 0.0001.

**Figure 6 ijms-23-08654-f006:**
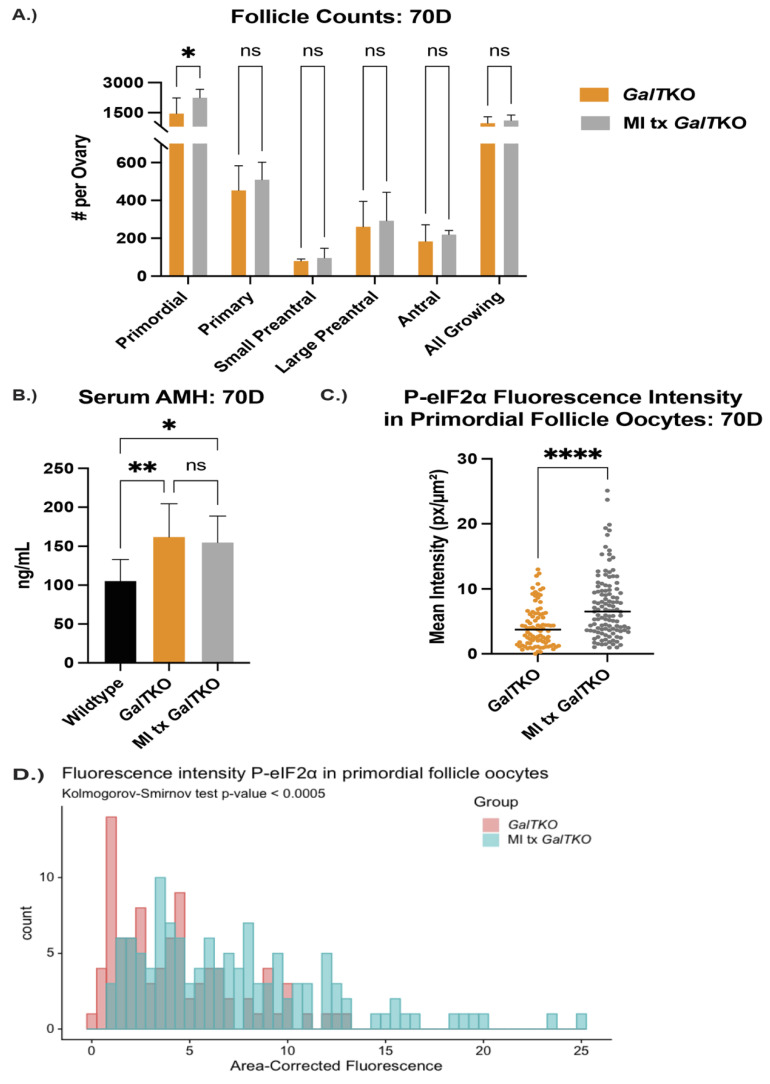
The effects of *myo*-inositol (MI) supplementation on the ovary. Follicle counts (**A**) comparing untreated *GalT*KO mice to MI supplemented *GalT*KO mice at 70D using two-way ANOVA. (**B**) Levels of serum AMH between WT and either untreated or MI supplemented *GalT*KO mice (one-way ANOVA). Median fluorescence intensity of P-eIF2ɑ in primordial oocytes is shown in (**C**), while the distribution of primordial oocyte staining intensity is depicted in (**D**) (KS test). A representative image of ɣ-H2AX staining in secondary follicle granulosa cells (**E**) and the number of secondary granulosa cells staining positive for ɣ-H2AX (**F**). White dashed circles denote secondary follicle boundaries. * *p* ≤ 0.05, ** *p* ≤ 0.01, **** *p* ≤ 0.0001.

**Figure 7 ijms-23-08654-f007:**
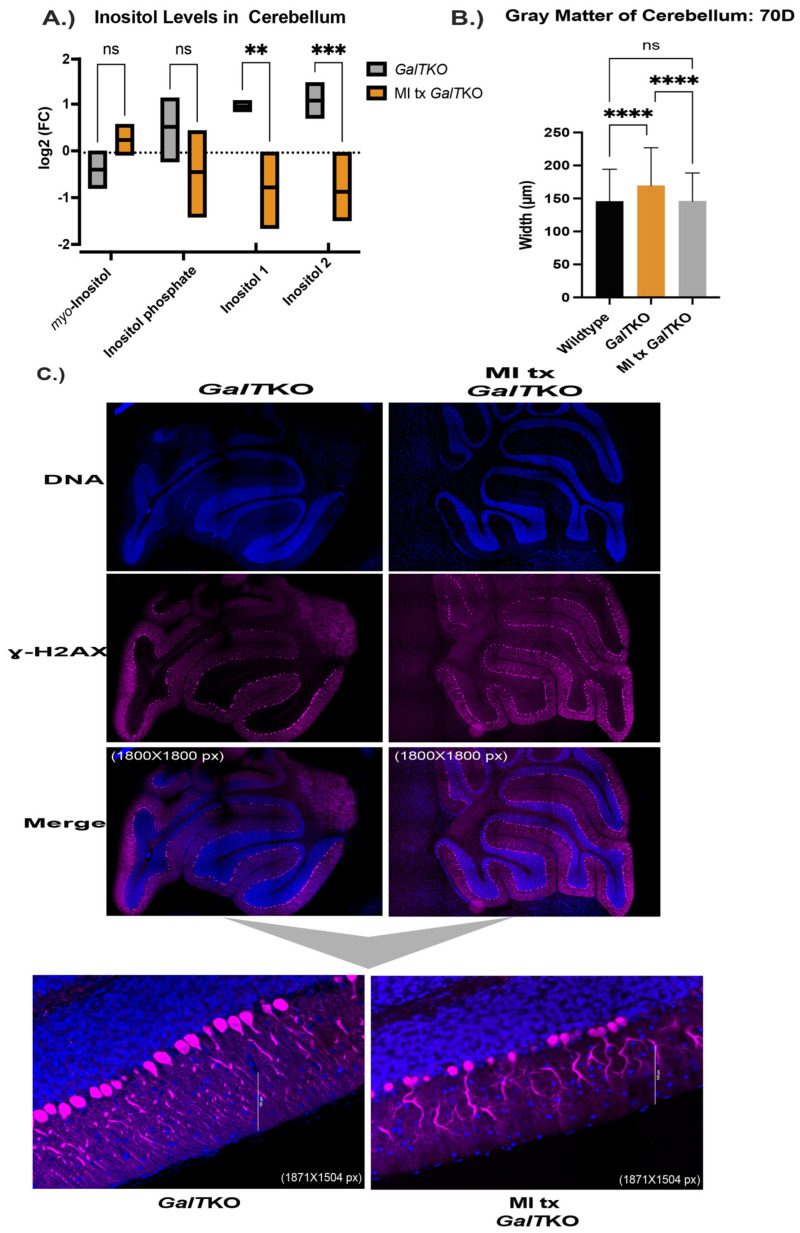
Characterization of MI supplementation in the brain and liver of *GalT*KO mice. Box and violin plot with line at the mean depicting measured inositol compounds between untreated *GalT*KO and MI treated *GalT*KO mice in the cerebellum (**A**). Gray matter, or combined widths of the molecular and granular layers in the brain, between WT, *GalT*KO, and MI treated *GalT*KO mice (**B**). Representative images of untreated and MI supplemented *GalT*KO cerebellum, immunostained for calbindin (**C**). The first six panels in (**C**) show the cerebellum at 1800 × 1800 pixels with subfigures at a magnification of 1857 × 1504 pixels. (**D**) H&E-stained liver histology with the larger panels at 10× and subfigures at 40×. Black arrows depict areas in the cytoplasm of hepatocytes with balloon-cell change. ** *p* ≤ 0.01, *** *p* ≤ 0.001, **** *p* ≤ 0.0001.

## Data Availability

Not applicable (no public database), but original data available when requested.

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
