# Peer review of "Harnessing the Power of Purple Sweet Potato Color and Myo-Inositol to Treat Classic Galactosemia"

_ijms, 2022, doi:10.3390/ijms23158654_

Round 1

Reviewer 1 Report

Authors provided sufficient feedback on the doubts I've raised regarding their paper. For this reason I believe that the paper is now ok to be published and I am not adding any further topics for discussion.

Author Response

Dear Reviewer 1,

  We appreciate your responses and critiques of the manuscript and feel your recommendations and the additional experiment we provided strengthen the manuscript immensely.  Thank you again for taking the time to review our manuscript and for your final comments. 

Best,

Synneva Hagen-Lillevik, MS, PhD

Reviewer 2 Report

The manuscript by Hagen-Lillevik et al., entitled “Harnessing the Power of Natural Remedies to Treat Classic Galactosemia” reports the supplementation with purple sweet potato color (PSPC) and myo-inositol (MI) could ameliorate dysregulation in the ovary, brain, and liver of our mutant mice. It is nicely written manuscript and can be accepted after incorporating the following corrections:

Figure 1 and 2 seem to be blurry and authors should provide a new image with better clarity. Further, Figure 2 A) seem to show HPLC chromatogram of two standard. If that is the case an overlay of the extract should also be shown. Title maybe misleading, hence, I recommend to change the “Natural Remedies” from title to “PSPC and MI supplementation”.

Author Response

Dear Reviewer 1,

Thank you for your insights.  Our responses are below.

-The manuscript by Hagen-Lillevik et al., entitled “Harnessing the Power of Natural Remedies to Treat Classic Galactosemia” reports the supplementation with purple sweet potato color (PSPC) and myo-inositol (MI) could ameliorate dysregulation in the ovary, brain, and liver of our mutant mice. It is nicely written manuscript and can be accepted after incorporating the following corrections:

-Figure 1 and 2 seem to be blurry and authors should provide a new image with better clarity.

RESPONSE: Thank you for letting us know these two images are blurry.  We have downloaded a JPG version that is 600 DPI for Figure 1 to correct this.  For Figure 2, it has been saved as a pdf with high quality.  Additionally, we will re-submit the Illustrator files for all figures in case the compression causes an issue.

-Further, Figure 2 A) seem to show HPLC chromatogram of two standard. If that is the case an overlay of the extract should also be shown.

RESPONSE: In Figure 2A, blue shaded rectangles have been added to show where the two standards fell, while peaked lines represent the peaks in our PSPC sample.  Additionally, to clarify the figure, the caption now reads (Lines 151-168):

Figure 2. The presence of anthocyanidin compounds and bioactive properties of the purple sweet potato color (PSPC). Peak values of present anthocyanidin backbones are shown in (A), with blue shaded regions indicating the standard peaks. Fold change in reactive oxygen species compared to untreated GalTKO fibroblasts (B).  Bars represent standard deviation in replicates.  Positive control (TBHP), 0.1% PSPC, and increasing concentrations of citric acid (0.26 mM, 0.52 mM, and 1.04 mM ) to account for the citric acid present in the PSPC extract.”

-Title maybe misleading, hence, I recommend to change the “Natural Remedies” from title to “PSPC and MI supplementation”.

RESPONSE:  This is a pertinent observation by the reviewer.  We had discussed this question previously, and we agree with the reviewer that it is misleading.  The title has now been changed to “Harnessing the Power of Purple Sweet Potato Color and Myo-Inositol to Treat Classic Galactosemia.”

Thank you for taking your time to review our manuscript.  We appreciate your comments and feedback.

Sincerely,

Synneva Hagen-Lillevik, MS, PhD

This manuscript is a resubmission of an earlier submission. The following is a list of the peer review reports and author responses from that submission.

Round 1

Reviewer 1 Report

In their manuscript, Hagen-Lillevik et al. investigate the effects of two supplements on a mouse model of classic galactosemia (GC) with a special focus on the central nervous and reproductive system as well as the liver. While exhibiting a sound methodology, some inheret aspects of the trial design need further clarification before the results can be published in the IJMS. 

Importantly, the used formulation of PSPC contains significant amounts of citric acid (CA), in itself an antioxidant. Therefore, the effects of PSPC on ROS generation and downstream consequences cannot be evaluated in the current experimental setup since effects of CA cannot be distinguished from those of PSPC. To rule out any confounding effect, experiments would have to be replicated in three groups, specifically untreated, treated with CA alone, and treated with PSPC (+ CA). 

In addition, the contents of PSPC are poorly characterized. A more detailed avaluation of  the specific anthrocyans as well as cross-batch variability would be beneficial to the interpretation of the results. Ideally, individuals anthrocyans could be isolated and identified as the active compounds, leading to more efficacious treatment of GC.

In addition, several minor issues/typos need to be addressed.

Methods:
It would be beneficial to the manuscript to include the dosage calculations mentioned in ll. 406+407 in the supplement. 

l. 61 : glycosylation instead of glucosylation

l. 72 : ovaries

Author Response

Dear Reviewer 1,

We really appreciate you taking the time to review our manuscript.  Your insights are spot on and valuable.  Please see our responses in blue below. 

-In their manuscript, Hagen-Lillevik et al. investigate the effects of two supplements on a mouse model of classic galactosemia (GC) with a special focus on the central nervous and reproductive system as well as the liver. While exhibiting a sound methodology, some inheret aspects of the trial design need further clarification before the results can be published in the IJMS. 

Importantly, the used formulation of PSPC contains significant amounts of citric acid (CA), in itself an antioxidant. Therefore, the effects of PSPC on ROS generation and downstream consequences cannot be evaluated in the current experimental setup since effects of CA cannot be distinguished from those of PSPC. To rule out any confounding effect, experiments would have to be replicated in three groups, specifically untreated, treated with CA alone, and treated with PSPC (+ CA). 

Thank you for your keen observation on this confounding ingredient.  We believe citric acid was used to make the liquid purple extract into a powder by the company who performed the extraction.  There appears to be few studies using citric acid as a dietary supplement for its antioxidant effects; however, we found an in vivo study that the authors showed that a minimum of 1-2 g/kg was required to induce a positive effect on reducing oxidative stress in the brain and liver of treated mice.  In the custom diet made for our study, the ratio of anthocyanins to citric acid is 10:1, and the amount of PSPC included in was 0.28% (by weight), which translated to 0.028% citric acid (by weight).  We calculated that a 20 g mouse will consume a maximum of 5 g of food per day, which would be 0.07g/kg of citric acid.  Such dose was 14-fold less than the minimum dose required to show an effect in the above-mentioned study (Abdel-Salam OM, Youness ER, Mohammed NA, Morsy SM, Omara EA, Sleem AA. Citric acid effects on brain and liver oxidative stress in lipopolysaccharide-treated mice. J Med Food. 2014 May;17(5):588-98. doi: 10.1089/jmf.2013.0065. Epub 2014 Jan 16. PMID: 24433072; PMCID: PMC4026104.).  Therefore, we do not think that the citric acid contained in the diet contributed much to the overall anti-oxidant effects.

In addition, in our fibroblast ROS studies, we calculate the concentration of citric acid added to be 5.2 mM and the anthocyanin concentration to be 26.1 mM, which is a much greater proportion of “antioxidant” material.  If needed, we can repeat the experiment using just CA in the concentration of the PSPC extract.  However, to do this would miss the deadline the editor has imposed for the revision as we would need to thaw and propagate fibroblasts for the experiments.  We would pursue this and include such data in the future publications.

In addition, the contents of PSPC are poorly characterized. A more detailed avaluation of the specific anthrocyans as well as cross-batch variability would be beneficial to the interpretation of the results. Ideally, individuals anthrocyans could be isolated and identified as the active compounds, leading to more efficacious treatment of GC.

This is an important issue that we have been working diligently to characterize the contents better.  The company (Food Ingredient Solutions) that extracted the color from the purple sweet potatoes claimed to have measured anthocyanidins through a company called Brunswick Laboratories, Inc. (MA, USA).  However, they would not give us the report, even when we asked to purchase it.  We tried to contact Brunswick Laboratories directly, but they have since gone out of business.  We also contacted the commercial labs Creative Proteomics (NY, USA) and IEH Laboratories and Consulting Group (WA, USA).  Creative Proteomics could only analyze 10+ samples and IEH did not have an anthocyanin/anthocyanidin panel.  As a research lab, we were unable to fund 10+ samples (and only had one) at Creative Proteomics.  Thus, we finally found Jes Persinger at the University of Colorado Boulder (CO, USA) (a co-author) who was able to analyze for the presence of anthocyanidins in our sample using HPLC.  Unfortunately, they are having problems with the machine and unable to finish what we planned in time.  Thus, the delphinidin and petunidin reported are peaks present based on a previous calibration of the machine.  Unfortunately, there appears to be a lack of laboratories able to perform the measurement of the anthocyanins/anthocyanidins in the United States.

We will continue to find someone to isolate specific anthocyanins/anthocyanidins, and we will perform future studies supplementing our mice (and hopefully one day humans) with individual anthocyanins.  However, individual anthocyanins are very difficult to purchase commercially for use in research studies, which is why we chose to have a custom extraction of the purple color, which is considered a pure source of “anthocyanins.”

In addition, several minor issues/typos need to be addressed.

Methods:
It would be beneficial to the manuscript to include the dosage calculations mentioned in ll. 406+407 in the supplement. 

This has been included in the methods section under “Custom Diets”

  1. 61 : glycosylation instead of glucosylation

This has been changed in the manuscript.

  1. 72 : ovaries

This has also been addressed in the manuscript.

Again, thank you for your insights.  We hope we have addressed them here and welcome any follow up questions or suggestions.

Sincerely,

Synneva Hagen-Lillevik and Kent Lai

Reviewer 2 Report

This paper aims to test if some supplements (PSPC and MI) may impact cellular pathways involved in CG.

Galactose metabolites are known to be the main cause of CG sequelae, thus to date main recognized intervention in GC patients is galactose dietary restriction even if this has not yet been proved to ameliorate long-term sequelae (which still need to clarified in underlying pathological mechanisms).

Main reported sequelae include growth restriction, progressive neurological damage and POI (which can link to other comorbidities such as CV aging, adrenal insufficiency, increased adiposity and decresaed BMD).

Recent study succeded in proving that there may be an impaired expression of the ISR pathway in the brain and ovary of GC mouse model linking to long-term sequelae, besides that there have yet not been any other study trying to understand if this can be limited by making interventions with supplements that can interfere with the ISR pathway.

This study show promising results but fails in:

- no statistical analysis methods have been reported and highlighted

- there is no report regarding materials and methods and how the study have precisely been performed

- more primordial follicles have been reported, but no differences in other types of follicles: does primordial follicles survive or not on a long term basis? "more" is not precise, how much is more? same for other part of the study, such as "more normal progression of estrus cycling then untreated" or "more intricate arborization", how much is more? what do you mean, give us some numbers

- when talking about diet, can you give us precise information about how this diet was intended?

- with regards to brain outcomes, how did you measure if changes in outcomes parameters have resulted in better neurological outcomes in terms of impairment? there is no functional study that can possibly assess if these changes on a cellular level can really have an outcome on a clinical standpoint

In conclusion, that anthocyanins have strong antioxidant and anti-inflammatory properties is already known and that this may link to positiv results in patients with ISR impairment may be expected. Besides that, this study fails to support on a clinical standpoint that reported results may really link to long term benefits.

Author Response

Dear Reviewer 2,

Thank you for taking the time to review our manuscript.  Your insights are greatly appreciated and helpful.  We will address your specific comments below in blue and in the manuscript.

This study show promising results but fails in:

- no statistical analysis methods have been reported and highlighted

Please refer to lines 674-682 for an overview of the statistical analyses uses.  For each section results, the statistical test and p values are included (Results section, beginning line 116) for each part of the figure.

- there is no report regarding materials and methods and how the study have precisely been performed

Please refer to lines 472-682 for the Materials and Methods section.  Additionally, brief methods are described in each of the results sections (beginning line 116).  Please let us know if one area in particular needs more clarification.

- more primordial follicles have been reported, but no differences in other types of follicles: does primordial follicles survive or not on a long term basis? "more" is not precise, how much is more? same for other part of the study, such as "more normal progression of estrus cycling then untreated" or "more intricate arborization", how much is more? what do you mean, give us some numbers

We have added a short paragraph explaining “folliculogenesis” and why more primordial follicles are important for the health and longevity of the ovary (370-377).  Also, we have included the actual numbers shown in the charts for the primordial follicles on line 166, 263-264, and the number of cells staining positive for y-H2AX on lines 177-178.  For the brain histology, we have taken out the wording of “more” and “increased” as we did not quantify the branches of the dendritic arbors but evaluated in a qualitative manner.  As this was a pilot study to see if supplementation could elicit any histological changes, we will include the quantification of the differences in the brain (3D staining and imaging) in our next publication. 

- when talking about diet, can you give us precise information about how this diet was intended?

We added a clarification of “daily supplementation” to line 75.  The mice consumed the supplements ad libitumas part of a custom diet made for this study.  We have also added the human equivalent dose and the estimations for how much of each supplement the mice received (Lines 490-493).

- with regards to brain outcomes, how did you measure if changes in outcomes parameters have resulted in better neurological outcomes in terms of impairment? there is no functional study that can possibly assess if these changes on a cellular level can really have an outcome on a clinical standpoint

Thank you for your insights regarding this important question.  We are aware that we do not report functional studies that illustrate any neurological phenotypic changes in our treated mice.  These studies are underway (Morris Water Maze and open-field test), which the results will give us a better idea of the clinical implications of these treatments.  The aim of this study was a pilot analysis of the histology to warrant the additional functional studies. 

In conclusion, that anthocyanins have strong antioxidant and anti-inflammatory properties is already known and that this may link to positiv results in patients with ISR impairment may be expected. Besides that, this study fails to support on a clinical standpoint that reported results may really link to long term benefits.

            We agree with the reviewer that the anti-inflammatory and antioxidant properties of anthocyanins are known facts, but we were not sure how much a role ISR plays in the POI phenotype of Galactosemia where other pathogenic factors beyond oxidative/ER stress exist.  Only through the experiments described in this study can we show the extent of involvement of ISR in the subfertility phenotype of our mouse model.

While there were not many functional studies reported here except the breeding trial after PSPC supplementation, our histological findings (ISR molecular markers and follicles counts) are congruent with previous studies from our lab using Salubrinal (citation below) to improve fertility and fecundity in our mouse model.  We acknowledge that functional studies are needed and human trials to determine the translation of our findings to human individuals with Classic Galactosemia.  We have shown some examples of how these two supplements may contribute positively to ovarian disease and in Classic Galactosemic-pathophysiology in particular (lines 350-365).  We have also added sentences on lines 459-463 regarding the application of the dosing used for the supplements in humans. 

Balakrishnan B, Siddiqi A, Mella J, Lupo A, Li E, Hollien J, Johnson J, Lai K. Salubrinal enhances eIF2α phosphorylation and improves fertility in a mouse model of Classic Galactosemia. Biochim Biophys Acta Mol Basis Dis. 2019 Nov 1;1865(11):165516. Doi: 10.1016/j.bbadis.2019.07.010. Epub 2019 Jul 27. PMID: 31362041; PMCID: PMC6751024.

Thank you again for your thoughts and insights.  Your suggestions have strengthened this manuscript.  We hope we have addressed the issues here and are open to more discussion and suggestions.

Sincerely,

Synneva Hagen-Lillevik and Kent Lai

Round 2

Reviewer 1 Report

I thank the authors for the explanation and clarifications supplied in their reply. However, I do not agree with their conclusion that the amount of citric acid can be considered to be irrelevant to the observed effects. Since the authors consider additional experiments not feasible in a timely manner, I recommend to reject the article in its current form.

Reviewer 2 Report

Dear authors,

thank you for claryfing and address better some of the issues which I've highlighted in my previous commentary.

I feel that all the changes amended are enough and agree with final publication.

Wish you best luck for further studies mainly on the functional side of the story!